

# Effects of outdoor ranging on external and internal health parameters for hens from different rearing enrichments

Md Saiful Bari[1,2], Yan C.S.M. Laurenson[2], Andrew M. Cohen-Barnhouse[1,2], Stephen W. Walkden-Brown[2] and Dana L.M. Campbell[1]

[1] Agriculture and Food, Commonwealth Scientific and Industrial Research Organisation (CSIRO), Armidale, New South Wales, Australia
[2] School of Environmental and Rural Science, University of New England, Armidale, New South Wales, Australia

Corresponding author
Dana L.M. Campbell,
Dana.Campbell@csiro.au

## ABSTRACT

In Australia, free-range layer pullets are typically reared indoors, but adult layers go outdoors, and this mismatch might reduce adaptation in laying environments. Enrichments during rearing may optimise pullet development and subsequent welfare as adult free-range hens. In the outdoor environment, hens may have greater opportunities for exercise and natural behaviours which might contribute to improved health and welfare. However, the outdoor environment may also result in potential exposure to parasites and pathogens. Individual variation in range use may thus dictate individual health and welfare. This study was conducted to evaluate whether adult hens varied in their external and internal health due to rearing enrichments and following variation in range use. A total of 1386 Hy-Line Brown® chicks were reared indoors across 16 weeks with three enrichment treatments including a control group with standard housing conditions, a novelty group providing novel objects that changed weekly, and a structural group with custom-designed structures to increase spatial navigation and perching. At 16 weeks of age the pullets were moved to a free-range system and housed in nine identical pens within their rearing treatments. All hens were leg-banded with microchips and daily ranging was assessed from 25 to 64 weeks via radio-frequency identification technology. At 64–65 weeks of age, 307 hens were selected based on their range use patterns across 54 days up to 64 weeks: indoor (no ranging), low outdoor (1.4 h or less daily), and high outdoor (5.2–9 h daily). The external and internal health and welfare parameters were evaluated via external assessment of body weight, plumage, toenails, pecking wounds, illness, and post-mortem assessment of internal organs and keel bones including whole-body CT scanning for body composition. The control hens had the lowest feather coverage ($p < 0.0001$) and a higher number of comb wounds ($P = 0.03$) than the novelty hens. The high outdoor rangers had fewer comb wounds than the indoor hens ($P = 0.04$), the shortest toenails ($p < 0.0001$) and the most feather coverage ($p < 0.0001$), but lower body weight ($p < 0.0001$) than the indoor hens. High outdoor ranging decreased both body fat and muscle (both $p < 0.0001$). The novelty group had lower spleen weights than the control hens ($P = 0.01$) but neither group differed from the structural hens. The high outdoor hens showed the highest spleen ($P = 0.01$) and empty gizzard weights ($P = 0.04$). Both the rearing enrichments and ranging had no effect on keel bone damage (all $P \geq 0.19$). There were no significant interactions between rearing treatments and ranging patterns for any of the health and

![PeerJ]

welfare parameters measured in this study ($P \geq 0.07$). Overall, rearing enrichments had some effects on hen health and welfare at the later stages of the production cycle but subsequent range use patterns had the greatest impact.

## INTRODUCTION

Free-range egg production is prevalent within Australia, as consumers perceive free-range eggs to be healthier and tastier than caged eggs (*Bray & Ankeny, 2017*). The free-range system is also perceived to improve hen welfare (*Pettersson et al., 2016a*) as hens have the choice to move freely outdoors, are exposed to daylight, and have greater opportunities for exercise and natural behaviours which might contribute to improved health and welfare. However, free-range systems can also comparatively bring increased risk of disease (*Fossum et al., 2009*), heat stress (*Singh et al., 2017*), predation (*Bestman & Wagenaar, 2014*), parasites (*Permin et al., 1999*), vent-pecking (*Bestman & Wagenaar, 2014*), and mortality (*Bestman & Wagenaar, 2014*; *Singh et al., 2017*; *Richards et al., 2012*). Furthermore, it is well documented both within Australia and internationally that range use varies by individual choice, with some hens ranging daily while others do not range at all (*Larsen et al., 2017*; *Pettersson, Freire & Nicol, 2016b*). This variation could result in large differences in activity and diet between hens which may impact health and welfare such as influencing body composition by lowering body fat accumulation (*Crespo & Esteve-Garcia, 2001*; *Renema et al., 1999*; *Sun et al., 2006*) and strengthening the bones (*Regmi et al., 2016*) and muscles (*Casey-Trott et al., 2017a*), although recent work with commercial free-range layers showed no relationship between range use and tibial bone strength (*Kolakshyapati et al., 2019*).

Range use has been shown to impact external welfare parameters of hens. Individually tracked free-range hens that use the outdoor area more, show less feather damage than hens that prefer to spend time indoors (*Mahboub, Müller & Borell, 2004*; *Rodriguez-Aurrekoetxea & Estevez, 2016*). Similarly, opportunistic scoring of hens on the range has shown better plumage condition compared with hens scored in the shed (*De Koning et al., 2018*) and better plumage in hens that ranged farther (*Chielo, Pike & Cooper, 2016*). Outdoor ranging hens or hens that have access to a range area show comparatively reduced footpad dermatitis (*Heerkens et al., 2016*; *Rodriguez-Aurrekoetxea & Estevez, 2016*) and range use keeps toenails shortened (*Campbell et al., 2017*; *Yilmaz Dikmen et al., 2016*). However, not all studies demonstrate strong relationships between individual ranging and welfare parameters. *Larsen et al. (2018)* found no association between range access and comb colour, beak, footpad, plumage, or keel bone condition, although hens that ranged further had better beak condition and darker comb colour. Ranging is related to hen body weight, but the direction of the relationship varies between studies rendering this relationship currently equivocal (*Campbell et al., 2016*; *Hartcher et al., 2016*; *Singh et al., 2016*).

Range use can also impact internal health and welfare parameters but data on individual patterns are currently limited. Ranging hens may have improved digestive and gut function over non-ranging hens as they ingest stones and grit that eventually contribute to heavier gizzards (*Singh et al., 2016*). Keel bone damage can reduce pop-hole usage, but the causal relationship is unclear (*Richards et al., 2012*). Furthermore, *Kolakshyapati et al. (2019)* recently showed no relationship between high and low range use and keel bone damage. Ranging hens might be susceptible to internal parasite infections such as *Ascaridia galli* that are present in soil (*Kaufmann et al., 2011*; *Permin et al., 1999*) as well as the range being contaminated by previous batches of hens (*Höglund & Jansson, 2011*). But some research shows a reduction in flock level parasitic infections with increased range use (*Sherwin et al., 2013*) as the outdoor hens excrete more in the range and less indoors thus lowering the density of faeces and possibility of reinfection (*Sherwin et al., 2013*). Necropsies of hens from varying housing systems in Sweden showed a higher occurrence of viral and bacterial infections and diseases in free-range and floor-based systems (*Fossum et al., 2009*). Overall, there is minimal information on both external and internal health and welfare parameters of individual free-range hens that vary in their range use patterns.

For optimising the health and welfare of adult hens, it has been suggested to provide similar environments for both the rearing and layer housing (*Janczak & Riber, 2015*). In Australian free-range systems (and elsewhere), the pullet rearing and adult housing environments are dissimilar. Adult hens range outdoors but the pullets are reared indoors, which could result in poorer adaptation of adults to the free-range environment. Although it is typically not feasible to provide outdoor access to pullets, rearing enrichments may be a method of improving the pullet's developmental environment (*Campbell, Haas & Lee, 2019*). Regularly changing novel objects might simulate the frequently changing and unpredictable free-range environment and improve hen's adaptation to stressful change as adults (*Campbell et al., 2018a*). Enriching with structures to enhance perching and encourage spatial navigation may increase physical development and spatial awareness (*Gunnarsson et al., 2000*) which could benefit ranging hens. Management approaches that include rearing enrichments in their pullet stages may improve adaptability, reduce stress, and improve hen immunity (*Arbona, Anderson & Hoffman, 2011*; *Moe et al., 2010*), thus reducing disease prevalence or infections in free-range hens.

The current study was conducted with the aim to evaluate the effect of individual ranging patterns on health parameters through post-mortem examination of free-range layers from different rearing enrichments. We predicted improved health and welfare of enriched over control hens and both benefits and consequences of ranging.

## MATERIALS & METHODS

### Ethical statement

All research was approved by the University of New England Animal Ethics Committee (AEC17-092).
## Animals and Housing

A total of 1386 Hy-Line® Brown layers were used for this study. The chicks and pullets were reared indoors at the Kirby poultry facility and the adults were housed in the Laureldale free-range facility at the University of New England, Armidale, NSW, Australia. Day-old chicks (including additional chicks delivered by the hatchery but not transferred to the laying facility) were reared across 16 weeks within 9 pens (6.2 m L × 3.2 m W) distributed across three separate rooms. The pullets were exposed to 3 separate rearing enrichment treatments. These included a control group having no extra materials over the pen standard of rice hulls as floor litter, a novelty group where novel objects were added and changed at weekly intervals (e.g., balls, bottles, bricks, brooms, brushes, buckets, containers, pet toys, plastic pipes, strings) and a structural group where four custom-designed H-shaped perching structures (L, W, H = 0.60 m) with two solid panels and one open-framed side were provided. Each of the rooms had one pen replicate per treatment, balanced for location within rooms. Shade cloth hung on the wire pen dividers visually isolating birds from each other. At 16 weeks of age, bird density was approximately 15 kg/m$^2$ or 9 pullets/m$^2$ (average 174 –190 pullets/pen). Round feeders provided *ad libitum* access to commercially-formulated mash appropriate for the developmental stage and nipples supplied *ad libitum* water access. These resources were provided as per the current Australian Model Code of Practice for the Welfare of Animals –Domestic Poultry (*Primary Industries Standing Committee, 2002*). Artificial lighting and temperature schedules followed the recommended Hy-Line® Brown alternative management guidelines (*Hy-Line, 2016*) but the LED lighting was maintained at 100 lux as the pullets were destined for outdoor access (no natural light was present during rearing). Mechanical ventilation with heating operated as needed but no cooling system was present. Chicks were infra-red beak-trimmed at the hatchery with a vaccination schedule as per regulatory requirements and standard recommendations including vaccination against Newcastle disease, Marek's disease, fowl pox, fowl cholera, egg drop syndrome, *Mycoplasma gallisepticum*, *Mycoplasma synoviae*, infectious bronchitis, infectious laryngotracheitis, and avian encephalomyelitis.

At 16 weeks, 1386 pullets were transferred to the Laureldale free-range facility and remixed within pen replicates (extra delivered chicks that were grown out were rehomed). The hens were housed within their rearing treatments across 9 pens located in a single shed ($n = 154$ hens/pen, indoor density approximately 9 hens/m$^2$). The indoor pens were visually separated via shade cloth and included nest boxes (2 small and 1 large nest box), perches, round hanging feeders and water nipples to fulfil the requirements of the Australian Model Code of Practice for the Welfare of Animals –Domestic Poultry (*Primary Industries Standing Committee, 2002*). Rice hulls were used as floor litter material with one complete litter replacement mid-way through the flock cycle. The LED lighting schedule gradually increased to 16 h light and 8 h dark by 30 weeks of age with an average pen intensity of 10.0 ($\pm$ 0.84 SE) lux (Lutron Light Meter, LX-112850; Lutron Electronic Enterprise CO., Ltd, Taipei, Taiwan) as measured at birds' eye height from 3 pen locations (front, middle, back) when the pop-holes were closed. This lux was the highest that could be achieved with the shed lighting system. The shed was fan-ventilated with no temperature or humidity control.

Each of the 9 pens was connected to an outdoor range area (31 m L × 3.6 m W for each pen, density approximately 1.4 hens/m$^2$) which was accessed via two pop-hole openings (18 cm W × 36 cm H). The range area immediately after the pop-holes was 1.1 m of concrete path, then 1.6 m of river rock followed by a grassed area with no additional trees or shelter. The grassed area became denuded following both hen access and the winter season. Each range was visually divided by shade cloth hung along the wire fences. Hens were provided access to the outdoor area from 25 weeks of age (May 2018) for most of the day time via automatic opening and closing of the pop-holes. The pop-holes opened at 9:15 am and closed after sunset daily. This equated to approximately 9 h of available ranging time across winter followed by approximately 11 h of available ranging time after daylight saving time started (October 2018).

### Radio-frequency identification (RFID) system

All birds were banded with microchips (Trovan® Unique ID 100 (FDX-A operating frequency 128 kHz) glued into adjustable leg bands (Roxan Developments Ltd, Selkirk, Scotland) to track their movement in and out of the range pop-holes via radio-frequency identification (RFID) systems. The RFID systems were designed and supported by Microchips Australia Pty Ltd (Keysborough, VIC, Australia) with equipment developed and manufactured by Dorset Identification B.V. (Aalten, the Netherlands) using Trovan® technology. For a schematic of the RFID system, see *Campbell, Horton & Hinch (2018b)*. The system recorded the date and time of each tagged bird passing through and in which direction (onto the range, or into the pen) with a precision of 0.024 s (maximum detection velocity 9.3 m/s). Individual ranging data were collected daily from 25 until 65 weeks of age.

### RFID data and selection of hens

RFID data from 56 until 64 weeks of age (54 days of data) were initially run through a custom-designed software program written in the 'Delphi' language (Bryce Little, Agriculture and Food, CSIRO, St Lucia, QLD, Australia) that filtered out any unpaired or 'false' readings that may occur if, for example, a hen sits inside the pop hole but does not complete a full transition onto the range or back into the pen. Once screened in this way, the data were used to select a sample of birds from each pen to conduct post-mortem examinations on. A total of 307 hens were selected across all of the 9 pens at 64 weeks of age. The selected hens were categorized as 'indoor'—accessed the outdoors on one or zero of the 54 days, 'low outdoor—accessed the range on 53 or 54 of the 54 days but only for 1 h 24 mins or less, and 'high outdoor'—accessed the range for 54 of 54 days for 5 h 12 mins to 9 h. Based on these criteria, a total of 95 indoor, 109 low outdoor and 104 high outdoor hens were selected from the flock. Maximum ranging times differed between pens, thus the high outdoor birds were selected as the highest for their pen, resulting in the variation in hours. Where possible, hen selection was balanced across pens within treatments, but some pen replicates did show higher numbers of the extreme ranging patterns than others.

## Pre-mortem welfare assessment

All the selected hens were weighed 4 to 5 days before dissection using electronic hanging scales (BAT1; VEIT Electronics, Moravany, Czech Republic). The external health and welfare parameters including feather loss at different body parts (neck, chest, back, wing, vent, tail) and footpad lesions were assessed using the scoring system described by *Tauson et al. (2005)*. In this scoring system, 4 scores were available for feather coverage with a score 4 indicating minimal feather damage, whereas a score 1 indicated bare skin. The back of the neck was scored separately from the front of the neck that was not included in the analyses as the majority of damage on the neck front was believed to have resulted from rubbing on the feeder rims rather than pecking. A maximum score of 24 could be obtained for feather condition across 6 body parts. Footpad lesions were scored as a 4 for a normal footpad with no lesions (an additional category to the *Tauson et al. (2005)* system) and a score 1 for swollen, infected bumble foot. The exact number of fresh or healing comb wounds were counted, and toenail length was measured in mm using a seamstress tape measure. Beaks were scored as 0, 1, or 2 indicating no, mild, or moderate damage respectively. The birds were also examined for any other external signs of injury or illness, such as body wounds, bleeding, abnormal respiratory sound etc., but none were observed in these hens. All selected hens were tagged with an additional leg band for identification in their home pen to enable later capture.

## Post-mortem health examination

Post-mortem examinations of the selected 307 hens were carried out across two days at 65 weeks of age at a separate post-mortem facility located 5.5km from the free-range facility. The selected hens of a single pen were transported to the post-mortem lab up to 2 h prior to the dissection using plastic carrier crates. Hens were killed using $CO_2$ to maintain hen physical structure for later scanning. Immediately after the $CO_2$ administration and cessation of all hen movement, the birds were opened by a veterinarian to examine the health condition of the hens (presence/absence of diseases) by inspecting the visceral organs for any abnormalities including haemorrhage, tumours, caseous necrosis and/or other exudates, the respiratory system (nares and trachea) for any haemorrhage, inflammation or exudates and the reproductive system for signs of salpingitis, being egg bound, or other abnormalities. Whether the hen was in production or not was determined by examination of the ovary and presence of active or regressing follicles. The spleen, gizzard, and right adrenal gland were removed and weighed (to the nearest mg) from each of the selected birds. The gizzard was emptied prior to weighing with surrounding connective tissue and fat removed. The jejunum, duodenum and ileum were opened longitudinally and the number of *Ascaridia galli* worms in each bird recorded. This is one of the most prevalent nematode parasites present within Australian commercial free-range systems (*Dao et al., 2018*). The carcass weight of each bird was taken after post-mortem examination (minus the removed organs and small intestines) prior to the Computed Tomography (CT) scanning.

## CT scanning and image analysis

Following post-mortem examination, all birds were scanned using a HiSpeed QX/I (2003) CT scanner (GE Medical Systems). Scans were performed using a voltage of 100 kV, a

current of 80 mA, a pitch of 0.75:1, a field of view of 500 × 500 mm, a thickness of 2.5 mm, and a spacing of 3.75 mm per rotation. Resultant images had a pixel matrix of 512 × 512, and hence a pixel resolution of 2.384 mm$^3$.

CT image stacks of each individual hen were analysed using ImageJ (Rasband W.S., ImageJ, US National Institutes of Health, Bethesda, Maryland, USA, https://imagej.nih.gov/ij/, 1887–2018) to generate pixel frequency plots against greyscale pixel values (0–255). Greyscale value density was defined using a linear regression parameterized to known carcass weights (g) recorded immediately prior to scanning. As such, the linear regression between density ($\rho$, $\mu$g/mm$^3$) and greyscale value (x) was given as:

$$\rho = 10.19(\pm 0.4546\sigma) \cdot x \tag{1}$$

where $\sigma$ is a standard deviation.

The weight of individual chicken (w, g) was therefore given as:

$$w = \sum_{x=0}^{255} y \cdot \alpha \cdot \rho \cdot 1000000 \tag{2}$$

where y is the frequency of each greyscale value; $\alpha$ is the pixel resolution (2.384 mm$^3$); and $\rho$ is density ($\mu$g/mm$^3$, Eq. (1)).

Body composition of each chicken was estimated assuming that the density of raw chicken fat was $\leq$865.64 $\mu$g/mm$^3$ (https://www.aqua-calc.com/page/density-table/substance/chicken-coma-and-blank-broilers-blank-or-blank-fryers-coma-and-blank-separable-blank-fat-coma-and-blank-raw), and that bone was identified as having a greyscale value of 255 (2598.45 $\mu$g $\pm$ 0.4546 $\sigma$), with the remaining mass being attributed to muscle. Some hens contained an egg which was not removed prior to scanning and was classified the same as bone in the images. All hens with an egg present (1/0) were later identified ($n = 87$) and removed from the analysis.

## Assessment of the keel bone

Following CT scanning, the keel bone was excised from each hen to assess deformities and damage. All keel bones were stored at $-20\,^\circ$C until thawed for processing. For processing, the fleshes on the keels were removed using a knife and scissors. The defects on the keels were examined by two experimenters using a visual scoring system for bending of the spine that was classified as low, moderate, or high based on the extent (Fig. 1). The two observers scored each bone independently and then immediately confirmed scores to ensure agreement for each bone. The dorsal surface of the tip of the keels were also observed for the presence of calluses to indicate healed fractures and the number of calluses were counted and recorded by two observers in agreement (Fig. 2). No fresh pre-mortem fractures were observed.

## Data and statistical analyses

Statistical analyses were conducted in JMP$^\circledR$ 14.0 (SAS Institute, Cary, NC, USA) with $\alpha$ set at 0.05. Data were transformed where needed but the raw values are presented in the tables and graphs. Non-significant interactions were removed from the final models and post-hoc

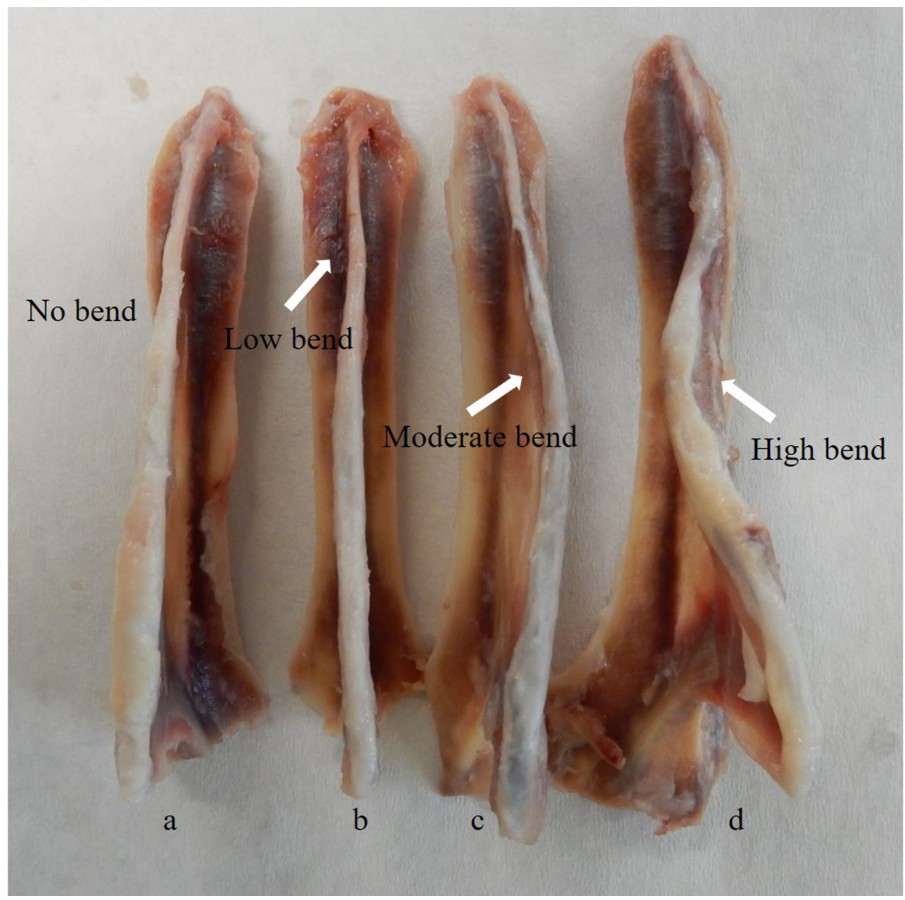

**Figure 1** **Different types of spine bending of keel bones.** Keel (A) indicates a spine with no bending, (B) low bending, (C) moderate bending and (D) indicates high bending in the spine. The white arrows indicate the specific part of bending on the spine.

Student's t-tests were applied to the least-squares means where significant differences were present. For the pre-mortem welfare assessment data, the per hen values ($n = 307$ hens) for the total feather score (up to 24) and the number of comb wounds were square-root transformed. The body weight and toenail length measurements per bird were compiled. General Linear Mixed Models (GLMM) were fitted to each welfare variable with rearing enrichment treatments and ranging patterns as fixed effects including their interaction and bird ID nested within pen, rearing treatment, and ranging as a random effect. Restricted maximum likelihood estimation methods were applied.

The feather scores for each body part and footpad damage scores were compiled per individual hen and the effects of rearing treatment, ranging, and their interaction were tested using ordinal logistic regressions.

The body composition data (muscle, fat, bone in grams) from the CT scans ($n = 307$), post-mortem parameters including organ weights (liver, spleen, right adrenal gland, and empty gizzard proportional to live body weight), and *A. galli* counts were compiled per individual bird ($n = 306$ hens, data from 1 bird were missing). The relative organ weight

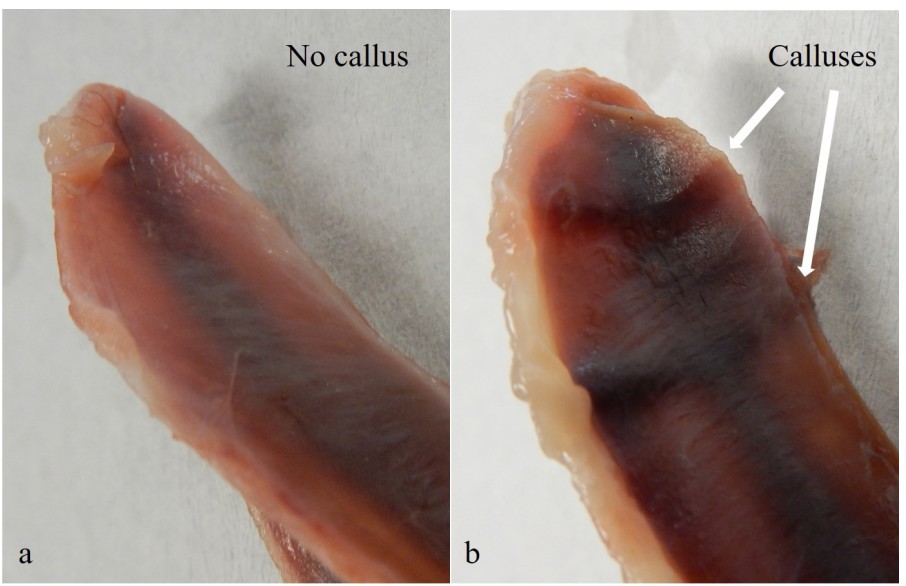

**Figure 2 Calluses on the dorsal surface of keel bones.** (A) indicates a keel surface with no calluses, but (B) shows two calluses as indicated by the white arrows.

proportional data were logit-transformed for analysis but expressed as percentages in the table. The *A. galli* count data were square-root transformed. GLMMs were fitted to test the fixed effects of rearing treatment and ranging groups including their interaction, with bird ID nested within pen, rearing treatment, and ranging as a random effect. The bone composition data of hens with an egg present ($n = 87$) were excluded from the analysis and data from 220 hens only were used.

Although a total of 307 birds were dissected, only 300 keels were assessed due to labelling issues with 7 keels. The qualitative data for the keel bones were compiled per individual bird ($n = 300$) including the overall presence of damage to the keel bone (yes/no), the type of keel spine bending, the presence (yes/no) and the number of calluses on the tips of the keels. Pearson's Chi-square tests for differences between rearing treatments and ranging patterns were conducted separately for each variable of each set of data. When conducting the Chi-square test with one treatment, the other treatment effect was blocked for each of the variables. The counts of calluses were square-root transformed and analysed using a GLMM to test the effects of rearing treatment and ranging groups including their interaction, with bird ID nested within pen, rearing treatment, and ranging as a random effect.

## RESULTS

### Pre-mortem welfare assessment

Enriched rearing treatments did not affect the live weight of hens ($F_{2,302} = 0.11$, $P = 0.90$) but the high outdoor hens had the lowest body weight ($F_{(2,302)} = 10.90$, $P < 0.0001$, Table 1). The total feather score of hens was the lowest in the control rearing treatment

**Table 1** **The welfare parameters of free-range hens at 64 weeks of age.** The least squares means ± standard error of the mean are presented for hens from different rearing treatments (control, novelty, structural) and ranging patterns (indoor, low outdoor, high outdoor).

| Variable | Category | Live weight (kg) | Feather score (out of 24) | Number of comb wounds | Nail length (cm) |
|---|---|---|---|---|---|
| Rearing enrichments | Control | 2.01 ± 0.02 | 21.31 ± 0.15[b] | 0.91 ± 0.13[a] | 1.54 ± 0.02 |
| | Novelty | 2.01 ± 0.02 | 22.83 ± 0.14[a] | 0.48 ± 0.12[b] | 1.58 ± 0.02 |
| | Structural | 2.02 ± 0.02 | 22.57 ± 0.14[a] | 0.68 ± 0.13[a,b] | 1.53 ± 0.02 |
| | P –value | 0.90 | <0.0001 | 0.03 | 0.12 |
| Ranging | Indoor | 2.05 ± 0.02[a] | 21.78 ± 0.15[b] | 0.79 ± 0.13[a] | 1.67 ± 0.02[a] |
| | Low outdoor | 2.04 ± 0.02[a] | 21.99 ± 0.14[b] | 0.82 ± 0.12[a,b] | 1.61 ± 0.02[b] |
| | High outdoor | 1.95 ± 0.02[b] | 22.94 ± 0.14[a] | 0.47 ± 0.12[b] | 1.37 ± 0.02[c] |
| | P –value | <0.0001 | <0.0001 | 0.04 | <0.0001 |

**Notes.**
[a,b]Dissimilar superscript letters indicate significant differences between rearing enrichments or ranging patterns ($P < 0.05$).
Raw values are presented with the analyses conducted on transformed data.

($F_{(2,302)} = 31.41$, $P < 0.0001$) indicating the most feather loss in these birds compared with the enriched hens (Table 1). The total feather score was highest in the high outdoor hens ($F_{(2,302)} = 18.06$, $P < 0.0001$) compared with low outdoor and indoor hens indicating the least feather loss in these birds (Table 1). The control hens had more comb wounds than the novelty hens but neither group differed from the structural group ($F_{(2,302)} = 3.63$, $P = 0.03$, Table 1). The high outdoor hens had fewer comb wounds compared with the indoor hens but neither ranging group differed from the low outdoor hens ($F_{(2,302)} = 3.08$, $P = 0.04$, Table 1). Rearing treatment did not affect toenail length ($F_{(2,302)} = 2.19$, $P = 0.11$), but the indoor hens had the longest nails and the high outdoor hens had the shortest ($F_{(2,302)} = 86.84$, $P < 0.0001$, Table 1). There was no interaction between rearing treatment and ranging patterns on any of the pre-mortem variables ($P \geq 0.11$).

A higher percentage (95%) of hens from the structural group showed feet with no damage (score 4) compared with the control (93%) and novelty groups (93%). A higher percentage of hens (97%) from the low outdoor ranging groups showed no footpad damage (score 4) compared with the high outdoor (88%) and indoor hens (95%). Only 0.92% of hens from the novelty group and 0.96% hens from the high outdoor ranging group showed the worst footpad condition (score 1). Ordinal logistic regression analysis showed ranging patterns significantly affected the footpad damage of hens ($\chi 2 = 7.07$, $df = 2$, $P = 0.03$) with high ranging increasing damage but the rearing treatments did not ($\chi 2 = 0.86$, $df = 2$, $P = 0.65$). There was no interaction between rearing treatment and ranging patterns ($P = 0.14$).

For other health issues examined pre-mortem, 0.01% of hens showed a prolapse. Most of the hens (91.86%) had no defects on the beaks with a scoring of '0', but 0.06% and 0.03% of hens had beaks with a score of '1' for mild and a score of '2' for moderate defects, respectively.

### Feather scores of individual hen body parts
Analyses of the feather scores of separate hen body parts showed rearing treatments had significant effects on plumage damage to the back of the neck, chest, and back with control hens showing the poorest feather coverage (Table 2). Ranging patterns had significant effects

on plumage damage to the chest with high outdoor hens showing the greatest plumage coverage (Table 2). There were no significant interactions between rearing treatments and ranging groups (all $P \geq 0.07$).

## Post-mortem parameters

Rearing treatments affected the relative weight of the spleen ($F_{(2,301)} = 4.82$, $P = 0.01$) with the spleens from the novelty hens having lower weight than the spleens from the control hens but neither group differed from the structural group (Table 3). The high outdoor birds' relative spleen weight was the highest ($F_{(2,301)} = 4.44$, $P = 0.01$). The high outdoor hens had higher empty gizzard weights than the indoor hens ($F_{(2,301)} = 3.22$, $P = 0.04$) but there was no effect of rearing treatment ($F_{(2,301)} = 0.85$, $P = 0.43$). Both the rearing treatments and ranging had no significant effects on the relative liver and adrenal weights (Table 3). There was no overall significant effect of rearing treatments on the number of *A. galli* worms in the GI tract ($F_{(2,301)} = 2.30$, $P = 0.10$), but post-hoc tests (which are more focussed to differentiate the clear visual differences in means, (*Hsu, 1996*) showed the novelty hens had more *A. galli* worms than the control hens and neither group differed from the structural hens (Table 3). There was no effect of ranging group on the number of *A. galli* worms ($F_{(2,301)} = 1.15$, $P = 0.32$). There were no significant interactions between rearing treatment and ranging for any of the measured variables (all $P \geq 0.21$).

Post-mortem examination also revealed all hens under study except one, were in production. There were no disease lesions observed on the respiratory system of the hens. One bird had a fatty liver, one peritonitis and one had keel adhesion. A cystic right oviduct was also present in 0.04% of hens.

## Body composition

There was a trend for an effect of rearing treatment on body fat ($F_{(2,302)} = 2.80$, $P = 0.06$) with the novelty group showing lower body fat (LSM ± SEM = 122.39 ± 1.78) than the control group (LSM ± SEM = 128.15 ± 1.91) but neither differed from the structural group (LSM ± SEM = 126.99 ± 1.83). Ranging had a significant effect on body fat ($F_{(2,302)} = 19.70$, $P < 0.0001$) and muscle ($F_{(2,302)} = 11.49$, $P < 0.0001$) with the high outdoor birds showing the lowest amount of fat and muscle (Fig. 3). There was no effect of rearing treatment on muscle mass ($F_{(2,302)} = 0.25$, $P = 0.78$). There was a trend for an effect of rearing treatments ($F_{(2,215)} = 2.84$, $P = 0.06$) on bone mass with the post-hoc tests showing the novelty hens had higher bone mass than the control group but the structural group did not differ significantly from either (Fig. 4). The ranging patterns had no significant effect ($F_{2,215} = 1.95$, $P = 0.14$) on bone mass of the hens. There were also no significant interactions of rearing treatments and ranging on any measured variable (all $P \geq 0.16$).

## Keel bone damage

The free-range hens from different rearing treatments and ranging patterns showed no significant differences in the overall presence of keel bone defects, presence of spine bending, spine bending types, and the presence of callus formation on the dorsal surface of keel tips (all $P \geq 0.19$, Table 4). The rearing treatments ($F_{(2,295)} = 0.40$, $P = 0.67$), ranging

**Table 2  The feather scores on different body parts for free-range hens.** The number and percentages of sampled hens within each group for each feather score category of six body parts (back of the neck, chest, back, wing, tail, vent) of free-range hens from different rearing treatments (control, novelty, structural) and ranging patterns (indoor, low outdoor, high outdoor) at 64 weeks of age. A score of 4 indicates the most feather coverage and a score of 1 the least, based on the scoring system of *Tauson et al. (2005)*. $N = 95$ for control, $N = 109$ novelty, $N = 103$ Structural, $N = 94$ indoor, $N = 109$ low outdoor and $N = 104$ for high outdoor groups. A '-' is given when no birds within any treatment group had that score.

| Feather location | Variable | Category | Damage score n (%) | | | | $\chi^2$, df, P |
|---|---|---|---|---|---|---|---|
| | | | 1 | 2 | 3 | 4 | |
| Neck (back only) | Rearing enrichments | Control | – | 14 (14.74) | 6 (6.32) | 75 (78.94) | 26.50, 2, <0.0001 |
| | | Novelty | – | 0 (0) | 6 (5.50) | 103 (94.5) | |
| | | Structural | – | 2 (1.94) | 0 (0) | 101 (98.06) | |
| | Ranging | Indoor | – | 8 (8.52) | 9 (9.57) | 77 (81.91) | * |
| | | Low outdoor | – | 8 (7.34) | 2 (1.83) | 99 (90.83) | |
| | | High outdoor | – | 0 (0) | 1 (0.96) | 103 (99.04) | |
| Chest | Rearing enrichments | Control | – | 49 (51.58) | 13 (13.68) | 33 (34.74) | 6.88, 2, 0.03 |
| | | Novelty | – | 36 (33.03) | 23 (21.1) | 50 (45.87) | |
| | | Structural | – | 37 (35.92) | 26 (25.25) | 40 (38.83) | |
| | Ranging | Indoor | – | 45 (47.87) | 20 (21.28) | 29 (30.85) | 33.26, 2, <0.0001 |
| | | Low outdoor | – | 58 (53.21) | 18 (16.51) | 33 (30.28) | |
| | | High outdoor | – | 19 (18.27) | 24 (23.08) | 61 (58.65) | |
| Back | Rearing enrichments | Control | – | 7 (7.37) | 27 (28.42) | 61 (64.21) | 55.43, 2, <0.0001 |
| | | Novelty | – | 0 (0) | 1 (0.92) | 108 (99.08) | |
| | | Structural | – | 1 (0.97) | 9 (8.74) | 93 (90.29) | |
| | Ranging | Indoor | – | 2 (2.13) | 11 (11.70) | 81 (86.17) | * |
| | | Low outdoor | – | 2 (1.83) | 17 (15.6) | 90 (82.57) | |
| | | High outdoor | – | 4 (3.85) | 9 (8.65) | 91 (87.5) | |
| Wing | Rearing enrichments | Control | – | – | 14 (14.74) | 81 (85.26) | 0.42, 2, 0.81 |
| | | Novelty | – | – | 14 (12.84) | 95 (87.16) | |
| | | Structural | – | – | 12 (11.65) | 91 (88.35) | |
| | Ranging | Indoor | – | – | 14 (14.89) | 80 (85.11) | 2.84, 2, 0.24 |
| | | Low outdoor | – | – | 17 (15.6) | 92 (84.4) | |
| | | High outdoor | – | – | 9 (8.65) | 95 (91.35) | |
| Tail | Rearing enrichments | Control | – | – | 19 (20.0) | 76 (80.0) | 4.70, 2, 0.10 |
| | | Novelty | – | – | 11 (10.09) | 98 (89.91) | |
| | | Structural | – | – | 13 (12.62) | 90 (87.38) | |
| | Ranging | Indoor | – | – | 19 (20.21) | 75 (79.79) | 5.13, 2, 0.08 |
| | | Low outdoor | – | – | 14 (12.84) | 95 (87.16) | |
| | | High outdoor | – | – | 10 (9.62) | 94 (90.38) | |
| Vent | Rearing enrichments | Control | 1 (1.06) | 13 (13.68) | 5 (5.26) | 76 (80.0) | * |
| | | Novelty | 0 (0) | 1 (0.92) | 0 (0) | 108 (99.08) | |
| | | Structural | 0 (0) | 0 (0) | 5 (4.85) | 98 (95.15) | |
| | Ranging | Indoor | 0 (0) | 6 (6.38) | 7 (7.45) | 81 (86.17) | * |
| | | Low outdoor | 1 (0.92) | 4 (3.67) | 2 (1.83) | 102 (93.58) | |
| | | High outdoor | 0 (0) | 4 (3.85) | 1 (0.96) | 99 (95.19) | |

**Notes.**
*Chi-square tests were not performed due to insufficient data within each scoring group.

Bari et al. (2020), *PeerJ*, DOI 10.7717/peerj.8720

**Table 3  The relative organ weights and *A. galli* worm counts of free-range hens.** The least squares means ± standard error of the mean of the percentages of relative organ weights and *A. galli* worm counts of free-range hens at 65 weeks of age from different rearing treatments (control, novelty, structural) and ranging patterns (indoor, low outdoor, high outdoor).

| Variable | Category | Liver weight (%) | Spleen weight (%) | Adrenal weight (%) | Gizzard weight (%) | *A. galli* in GI tract (N) |
|---|---|---|---|---|---|---|
| | Control | 2.59 ± 0.04 | 0.097 ± 0.002[a] | 0.003 ± 0.01 | 1.72 ± 0.03 | 4.52 ± 0.84 |
| Rearing enrichments | Novelty | 2.60 ± 0.04 | 0.089 ± 0.002[b] | 0.004 ± 0.01 | 1.67 ± 0.02 | 7.03 ± 0.79 |
| | Structural | 2.51 ± 0.04 | 0.092 ± 0.002[a,b] | 0.025 ± 0.01 | 1.69 ± 0.03 | 5.66 ± 0.80 |
| | Test statistics | $F_{(2,301)} = 2.23, P = 0.11$ | $F_{(2,301)} = 4.82, P = 0.01$ | $F_{(2,301)} = 1.82, P = 0.16$ | $F_{(2,301)} = 0.85, P = 0.43$ | $F_{(2,301)} = 2.30, P = 0.10$ |
| | Indoor | 2.58 ± 0.04 | 0.091 ± 0.002[b] | 0.004 ± 0.01 | 1.65 ± 0.03[b] | 4.88 ± 0.85 |
| Ranging | Low outdoor | 2.53 ± 0.04 | 0.089 ± 0.002[b] | 0.024 ± 0.01 | 1.69 ± 0.02[a,b] | 5.59 ± 0.78 |
| | High outdoor | 2.60 ± 0.04 | 0.097 ± 0.002[a] | 0.003 ± 0.01 | 1.74 ± 0.03[a] | 6.74 ± 0.80 |
| | Test statistics | $F_{(2,301)} = 1.29, P = 0.28$ | $F_{(2,301)} = 4.44, P = 0.01$ | $F_{(2,301)} = 0.57, P = 0.57$ | $F_{(2,301)} = 3.22, P = 0.04$ | $F_{(2,301)} = 1.15, P = 0.32$ |

**Notes.**

[a,b] Dissimilar superscript letters indicate significant differences between rearing enrichments or ranging patterns ($P < 0.05$). Raw values are presented with the analyses conducted on transformed data.

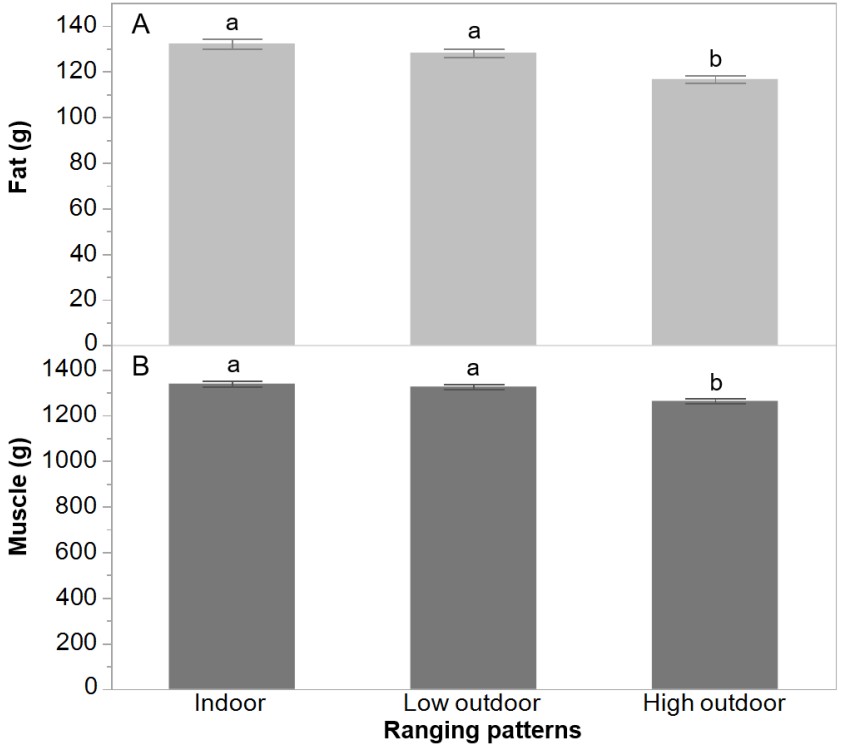

**Figure 3 The relative CT-scanned body composition of hens from different range use patterns.** The least squares means ± standard error of the mean of fat (A) and muscle (B) are presented from hens at 65 weeks of age that did not range (indoor), or ranged daily for low or high amounts of time. [a,b] Dissimilar superscript letters indicate significant differences between ranging patterns ($P < 0.0001$).

patterns ($F_{(2,295)} = 1.31$, $P = 0.27$) and their interaction ($P = 0.55$) had no effect on the number of calluses on the tip of the dorsal surface of the keels.

## DISCUSSION

This study assessed the effects of different rearing enrichments and outdoor ranging patterns on the external and internal health and welfare of free-range hens at 64 to 65 weeks of age. Hens enriched during rearing had better feather coverage than control hens and the hens enriched with novel objects had fewer comb wounds than the control hens. The novelty hens also had lower spleen weights than the control birds. Range access had multiple clear effects with the hens that ranged the most showing lower total body weight, lower fat and muscle content, better plumage, shorter toenails, the highest spleen weight and fewer comb wounds than the indoor birds. Rearing or ranging did not affect the prevalence and degree of keel bone damage, but most sampled birds showed some type of damage. Overall, enriched rearing environments still had some effects on hens at the later stages of the production cycle but subsequent range use patterns had the greatest impact.

Hens reared with enrichments had better plumage indicating a persistent effect of rearing conditions through until later in the production cycle. Multiple studies have documented the positive long-term effects of providing substrates during rearing where

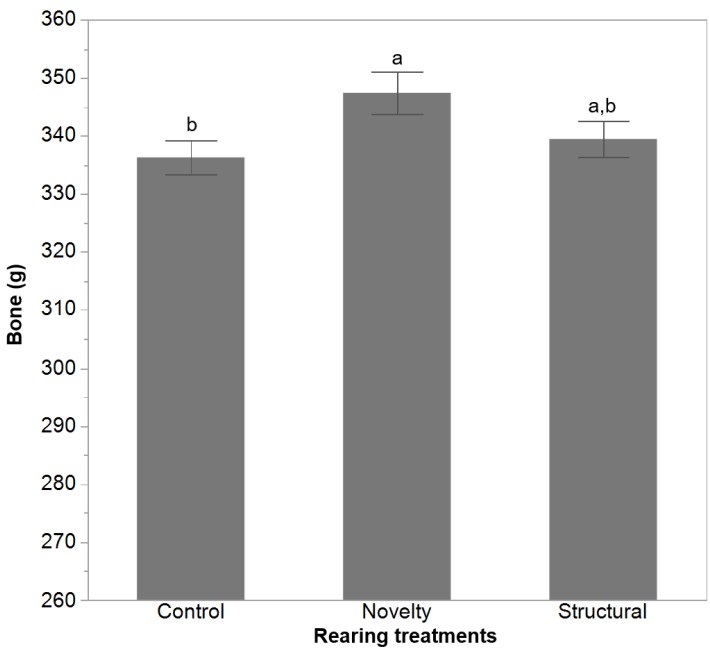

**Figure 4 CT-scanned bone mass of hens from different rearing treatments.** The least squares means ± standard error of the means are presented for free-range hens from control, novelty or structural rearing treatments at 65 weeks of age. [a,b] Dissimilar superscript letters indicate significant differences between rearing treatments as identified by a post-hoc Student's $t$-test. Only the data from the hens that had no eggs present during scanning were considered ($n = 220$).

opportunities to forage and dust bathe are suggested to prevent the development of feather pecking (reviewed in *Rodenburg et al., 2013*; *van de Weerd & Elson, 2006*) although substrate availability during the laying period is also critical (*Rodenburg et al., 2013*). All birds in this study were both reared and then housed with access to a floor litter substrate but the additional pen enrichments still had positive impacts. *Huber-Eicher & Audigé (1999)* did find that access to perches during rearing significantly reduced the risk of feather pecking. The structural rearing group had access to elevated perches for 16 weeks, and some of the objects initially placed in the novelty pens did allow chicks to perch (e.g., bricks, containers) with later objects provided to allow pecking/exploration (e.g., strings, pet toys). The increased complexity of the enriched pens may have improved behavioural development or provided more opportunities for the pullets to regulate their social interactions by having elevated escape areas. These positive effects of enrichments contrast with the findings by *Hartcher et al. (2015)* who reported that rearing enrichments of polypropylene pecking strings, whole oats and increased litter depth from 12 days of age had no effect on the plumage of adult free-range hens at 43 weeks of age. These discrepancies between results might have occurred due to the variation in novel objects of the different studies or age at first provision. Pullets did not show feather damage at the end of rearing (D Campbell, 2018, unpublished data) but it is unclear whether pecking behaviour in the control hens resulted from behavioural patterns established during rearing, and/or if the

adult control hens were more susceptible to environmental stress in the free-range setting which triggered the development of the negative pecking behaviour.

The novelty hens had lower relative spleen weights than the control hens but showed a higher worm count. Differences in spleen size can be related to parasite burden (*John, 1995*) where a greater burden increases spleen size. Spleen size can also reduce with stress where *Odihambo Mumma et al. (2006)* found that an adrenocorticotropin (ACTH) treatment to experimentally increase stress in layers reduced relative spleen weight compared with control hens. Thus, it is unclear the causes for the differences in spleen size in the current study. Additionally, only adult *A. galli* worms (no other infection life stages or other parasites) were counted in the current study and other measured organs were not affected by rearing treatments. Therefore, further study would be needed to confirm any relationship between rearing enrichments, stress, infections and spleen size in adult hens.

The outdoor hens had relatively heavier spleens. This heavier weight could be a result of reduced stress through outdoor access and potential behavioural freedom. However, experimental exposure to a lipopolysaccharide stressor from *E. coli* increased relative spleen weight and thus the outdoor hens may have had higher spleen weights through exposure to more pathogens (*Shini, Shini & Huff, 2009*). Increased spleen weight is a very general response to many types of infections although other factors might be involved (*Smith & Hunt, 2004*) such as seasonal influence (*John, 1994*). Outdoor hens were previously reported to be more stressed as indicated by the heterophil/lymphocyte ratio (*Mahboub, Müller & Borell, 2004*). However, variation in spleen weight might be dependent on type of stressors and further research is needed to confirm any relationship between ranging and spleens. There was no significant difference in the *A. galli* parasite burden between indoor and outdoor hens which is supported by recent findings by (*Bestman et al., 2019*) and previous research showing high parasite loads for hens housed in litter systems (*Permin et al., 1999*) suggesting all hens in the free-range system can become infected. However, further studies, particularly on commercial farms, are still required to confirm the individual-level relationship between parasitic burden and ranging patterns.

The hens that ranged the most showed the best plumage coverage which provides further support to previous research. Several studies have demonstrated better plumage in individuals, or flocks that show more frequent use of the outdoor range (*De Koning et al., 2018*; *Lambton et al., 2010*; *Mahboub, Müller & Borell, 2004*; *Rodriguez-Aurrekoetxea & Estevez, 2016*), or who range farthest when outdoors (*Chielo, Pike & Cooper, 2016*). It might be that hens outdoors are able to or are motivated to engage in more foraging compared with hens indoors (*Campbell et al., 2017*) where a lack of foraging is often redirected to feather pecking, causing plumage damage (*Bestman, Koene & Wagenaar, 2009*; *Gilani, Knowles & Nicol, 2013*; *Rodenburg et al., 2013*). This result contrasts with that of *Larsen et al. (2018)* who found no association between outdoor ranging and plumage condition of Hy-Line® Brown hens in commercial Australian conditions. It is also possible that better plumage coverage led to more ranging if hens were able to thermoregulate more effectively and/or avoid sun exposure on bare skin.

The outdoor rangers also had shorter toenails which confirms a previous finding at the same experimental research facility (*Campbell et al., 2017*) as well as research

Bari et al. (2020), *PeerJ*, DOI 10.7717/peerj.8720

**Table 4 Keel bone defects of free-range hens.** The number and percentages of keel bone damage of free-range hens at 65 weeks of age from different rearing treatments (control, novelty, structural) and ranging patterns (indoor, low outdoor, high outdoor). Values of each category of damages are presented as n (%) but the number of calluses on tips of keels are expressed as least squares means standard error of mean (LSM ±SEM). For the number of calluses on tips, raw values are presented with the analyses conducted on transformed data.

| Treatment | Category | N | Damages n (%) | Spine bending n (%) | Spine bending types n (%) | | | Callus formation n (%) | Number of calluses (LSM ± SEM) |
|---|---|---|---|---|---|---|---|---|---|
| | | | | | Low | Moderate | High | | |
| Rearing enrichments | Control | 91 | 71 (78.02) | 57 (62.64) | 46 (50.55) | 7 (7.69) | 4 (4.40) | 40 (43.96) | 0.71 ± 0.11 |
| | Novelty | 106 | 82 (77.36) | 68 (64.15) | 51 (48.11) | 7 (6.60) | 10 (9.43) | 42 (39.62) | 0.70 ± 0.10 |
| | Structural | 103 | 73 (70.87) | 66 (64.08) | 53 (51.46) | 8 (7.77) | 5 (4.85) | 39 (37.86) | 0.60 ± 0.10 |
| Test statistics, df, P | | | $\chi^2 = 1.69, 2, 0.43$ | $\chi^2 = 0.06, 2, 0.97$ | $\chi^2 = 2.79, 2, 0.84$ | | | $\chi^2 = 0.78, 2, 0.68$ | $F_{(2,295)} = 0.40, 0.67$ |
| Ranging patterns | Indoor | 92 | 68 (73.91) | 57 (61.96) | 42 (45.65) | 9 (9.78) | 6 (6.52) | 39 (42.39) | 0.72 ± 0.10 |
| | Low outdoor | 107 | 81 (75.70) | 71 (66.36) | 57 (53.27) | 8 (7.48) | 6 (5.61) | 36 (33.64) | 0.56 ± 0.10 |
| | High outdoor | 101 | 77 (76.24) | 63 (62.38) | 51 (50.50) | 5 (4.95) | 7 (6.93) | 46 (45.54) | 0.73 ± 0.10 |
| Test statistics, df, P | | | $\chi^2 = 0.15, 2, 0.93$ | $\chi^2 = 0.52, 2, 0.77$ | $\chi^2 = 2.60, 2, 0.86$ | | | $\chi^2 = 3.29, 2, 0.19$ | $F_{(2,295)} = 1.31, 0.27$ |

comparing between hens from free-range or caged systems (*Yilmaz Dikmen et al., 2016*). This is probably due to increased walking and scratching outside which allows hens to appropriately manage growing nail length.

Hens that ranged the most showed the lowest body weight, specifically, lower body fat and muscle, but not lower skeletal mass. The high outdoor hens showed an average body weight (1.95 kg) lower than the indoor hens but were within the limit of the expected body weight by breed standards (1.90 –2.02 kg; *Hy-Line, 2016*). Previous research has found some evidence of a similar negative relationship between body weight and range use, but not at all measured age points (*Campbell et al., 2017*) and some authors have found the opposite relationship (*Singh et al., 2016*). This negative relationship might be due to the ingestion of vegetation, insects or grit during ranging and thus consumption of less formulated food (*Singh & Cowieson, 2013*). This would also correspond with the higher empty gizzard weight observed in the ranging hens, similar to findings of larger gizzards in free-range versus caged hens (*Yang et al., 2014*). The reduction in body weight might also be a result of greater energy utilisation during locomotion, although greater exercise opportunities have previously been shown to increase bone and muscle development (*Casey-Trott et al., 2017a*; *Regmi et al., 2016*) which was not found in the outdoor hens. Other measured skeletal properties rather than overall mass may have revealed differences between ranging groups although recent work showed no effect of ranging on multiple tibial measurements across hens from a commercial aviary-free-range system (*Kolakshyapati et al., 2019*). There is currently limited knowledge in the comparative activity levels of hens that remain indoors, and this warrants further investigation.

The present study suggested no effect of rearing enrichments and ranging patterns on keel damage of free-range hens. This coincides with observations on commercial farms that assessed damage via both dissection (*Kolakshyapati et al., 2019*) and palpation (*Larsen et al., 2018*) and found no association between individual ranging and keel damage. It was expected that rearing with structural enrichments may have reduced the later occurrence of fractures (*Casey-Trott et al., 2017b*), but this was not supported by the current study. The actual relationship between range use and keel damage might be inconclusive because painful keel fractures might prevent birds passing through the pop holes (*Richards et al., 2012*) and ultimately reduces range access. The novelty rearing treatment increased bone mass relative to the control birds which may have been a result of novel objects that allowed perching in the first 2 weeks of life (e.g., overturned containers). The birds in the structural treatment were first observed starting to perch at 2 weeks of age onwards. Further specific bone measures would confirm any effects of the rearing treatments on skeletal development.

## CONCLUSION

The study showed that rearing enrichments had long-term effects on adult free-range hens, particularly in reducing the degree of plumage damage. However, subsequent individual ranging patterns by the hens had a stronger influence on their health and welfare with high outdoor use resulting in better plumage, fewer comb wounds, shorter nail length, higher

spleen and gizzard weight, but lower body weight, fat and muscle. Rearing enrichments are thus recommended for long-term positive effects on hen welfare, but management of range access may have the strongest welfare impact. This study was conducted in an experimental setting with small flock sizes and low incidence of infection. Large commercial groups of layers are likely to be exposed to more pathogens where outdoor access may have different effects on hen susceptibility. Similar long-term studies on commercial free-range farms would confirm the benefits and consequences of different ranging patterns.

## ACKNOWLEDGEMENTS

We would like to thank all staff and students of UNE and CSIRO who contributed in technical and husbandry assistance, and data collection for this study. We thank M. Raue (UNE) for assistance with the CT scanning. We also thank P. Scott and N. Fernando (Scolexia Pty. Ltd) for vaccination advice during pullet rearing.

### Funding

Poultry Hub Australia provided funding for the research (grant number 2017-20). Md Saiful Bari was supported by a University of New England and Commonwealth Scientific and Industrial Research Organisation (CSIRO) postgraduate scholarship. The funders had no role in study design, data collection and analysis, decision to publish, or preparation of the manuscript.

### Grant Disclosures

The following grant information was disclosed by the authors:
Poultry Hub Australia: 2017-20.
University of New England and Commonwealth Scientific Industrial Research Organisation (CSIRO) postgraduate scholarship.

### Competing Interests

The authors declare there are no competing interests.

### Author Contributions

- Md Saiful Bari and Dana L.M. Campbell conceived and designed the experiments, performed the experiments, analysed the data, prepared figures and/or tables, authored or reviewed drafts of the paper, and approved the final draft.
- Yan C.S.M. Laurenson performed the experiments, analysed the data, authored or reviewed drafts of the paper, and approved the final draft.
- Andrew M. Cohen-Barnhouse and Stephen W. Walkden-Brown conceived and designed the experiments, performed the experiments, authored or reviewed drafts of the paper, and approved the final draft.

## Animal Ethics

The following information was supplied relating to ethical approvals (i.e., approving body and any reference numbers):

The University of New England Animal Ethics Committee approved the research (AEC17-092).

## Data Availability

Data are available in the CSIRO Data Access Portal. Campbell, Dana; Bari, Saiful; Laurenson, Yan; Walkden-Brown, Stephen; Dyall, Timothy; Cohen-Barnhouse, Andrew (2019): Free-range layer health and welfare data. CSIRO. Data Collection. https://doi.org/10.25919/5da6aa06c7761.

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
