# Peer review of "Effects of outdoor ranging on external and internal health parameters for hens from different rearing enrichments"

_PeerJ, doi:10.7717/peerj.8720_

## Round 0.1 · original submission · Minor Revisions

Dear authors, the reviewers identified some minor points that should be addressed before acceptance of your article. We await your revised version.

Reviewer 1 ·

Basic reporting

No comment

Experimental design

The research is within the aim and the scope of the journal. The hypothesis is well explained and the findings and the conclusions are consistent with the aim.
The methods used are deeply explained

Validity of the findings

The results and discussion are well written apart from very minor issues.
line 406 and 410 there is a mistake in table numbering
line 468 and 472: In table 3 the worm count is higher in novelty hens than in control while in the text seems to be lower.
Along with the paper, there are minor editing mistakes (see the revision in the text.
In table n.3 it should be better to add the unit at the column Worms in GI tracts

Additional comments

No comment

Annotated reviews are not available for download in order to protect the identity of reviewers who chose to remain anonymous.

Reviewer 3 ·

Basic reporting

Is fine

Experimental design

Is fine

Validity of the findings

Is fine

Additional comments

This paper is very interesting because it looks at hen health and welfare related to range use and rearing conditions. Although several studies have been done on this topic, the authors used methods to track individual hens, which enables them to say much more about possible relationships on an individual level instead on a flock level. Furthermore, the paper is well written in good English, well structured and contains the necessary information, including some figures and tables. Therefore, my suggestions for changes are only minor.

Specific comments
L48 ‘greater exposure to parasites and pathogens’. Compared to what? To indoors? Are you sure outdoors is a greater exposure? When comparing outcomes of different studies on this topic (loose housing as such may be the risk factor and not the outdoor run per se), I would suggest to state it a bit more careful, for example ‘potential exposure to …’
L58 56 days or 54 days, as is written in L209?
L82-84 ‘increased risk of disease, heat stress, predation, parasites, vent-pecking, and mortality (Bestman & Wagenaar, 2014; Fossum et al. 2009; Lay et al. 2011; Richards et al. 2012; Singh et al., 2017)’ Which risk factor is mentioned by which author?
L94 ‘tracked free-range birds that use the outdoor area more, show less feather damage than hens that’. Add a ‘,’
L102 ‘Larsen et al. (2018) found no association between range access and comb color, beak, footpad ‘. Did Larsen et al investigate range access (a yes/no variable) or range use (with a degree of use)?
L111 ‘unclear Richards’. I guess a ‘.’is missing between these 2 words?
L112-114 ‘Ranging hens may be more susceptible to internal parasite infections such as Ascaridia galli (Kaufmann et al., 2011) as the range can be contaminated by previous batches (Höglund & Jansson, 2011)’. See my comment above (L48). Are you sure Kaufmann et al proved that it was the range or can they, based on their results and the references they use, only suggest/think that is was the outdoor run? I am asking this, because after I searched myself about the risk of the free-range concerning parasites, there seemed to be a lot of suggestion (copy pased by several authors) and nearly no proofs, ending up with the idea that it is not clear whether it is the loose housing or the range that causes the risk. I would suggest to chose a bit more careful phrasing here. The hen house can also be contaminated by previous batches.
L118 ‘of viral and bacterial diseases’: diseases or infections? Clinical or subclinical. Had the hens been ill or were only antibodies found without having been visibly ill?
L158 ‘bird density was approximately 15 kg/m2’. Can mention the density in hens/m² as well?
L165 Was there any daylight in the hen house? And what was the light/dark schedule?
L172 At which stocking density in hens/m²?
L203 ‘RFID system, see (Campbell et al., 2018b).’ Add a ‘,’ (2x)
L209 54 days. The abstract says 56 days.
L217-2018 1.4 hours, 5.2 hours: 1.4 and 5.2 is how many minutes on top of the 1 and 5 hours? Do you mean 0.4*60=24 and 0.2*60=12 minutes?
L239-240 ‘other external signs of injury or illness’. Can you give some examples, ‘such as …’
L247 ‘Hens were killed using CO2.’ Is this a standard method? Is decapitation no more animal-friendly than suffocation?
L264-265 ‘Following post-mortem examination all birds were CT scanned on the day of dissection. A total number of 307 Hy-Line® Brown layers were scanned using the HiSpeed QX/I (2003) CT scanner’. Remove the information that is already given elsewhere in your paper.
L277 Leave a white line above and below the formula
L280 Leave a white line above and below the formula
L277-287 Consider to limit the number of decimals. How relevant is for example .64 when talking about 865.64 μg/mm3? Consider to replace such numbers by 866 μg/mm3
L309 why square root transformed?
L320 ‘logit’ or ‘log’ and why transfomed?
L332 why square root transformed?
L354 and further Consider to use no or less decimals when presenting percentages, or only below a certain value. For example 95% instead of 95.15% and 93% instead of 92.6%. In case of % below for example 10% you can consider to use 1 or 2 decimals.
L355 was it on purpose not to show the feet data in a table?
L364-366 Was it on purpose not to show the other health issue in tables?
L384-386 ‘There was a trend for an effect of rearing treatments (F(2, 215) = 2.84, P = 0.06) on
bone mass with the post-hoc tests showing the novelty hens had higher bone mass than the control group but not the structural group did not differ significantly (Figure 4).’ Perhaps 1x ‘not’ should be removed?
L394 Table 3 instead of Table 4
L398 Table 3 instead of Table 4
L426 ‘the highest spleen weight, and fewer comb wounds’ Consider removing ‘,’
L477 ‘Further studies, particularly on commercial farms, are’ Add a ‘,’
L493 change ‘facility Campbell et al. (2017) as well’ into ‘facility (Campbell et al., 2017)
L501-505 Can more excercise and thus more energy burning from fat/muscle explain the negative relationship between range use and body weight?
L521 ‘al., 2017a), but this’. Add a ‘,’
L558 Change ‘Bestman M, Niekerk TV, Haas END, Ferrante’ into ‘Bestman M, Niekerk T van, Haas EN de, Ferrante’
L594 Change ‘De Koning C, Kitessa SM, Barekatain R, and Drake K.’ into ‘Koning C de, Kitessa SM, Barekatain R, and Drake K. and then place this reference between ‘Kolakshyapati et al 2019’ and ‘Lambton et al., 2010’.
L681 Change ‘Rodenburg TB, Van Krimpen MM, De Jong IC, De Haas EN, Kops’ into ‘Rodenburg TB, Krimpen MM van, Jong IC de, Haas EN de, Kops’
L711 Change ‘Van de Weerd HA, and Elson A.’ into ‘Weerd HA van de, and Elson A.’
Figure 4 Remove ‘,’ ‘from control, novelty, or structural’
Table 1 Replace and by , : (indoor, low outdoor, high outdoor)
Table 2 Is it possible to insert some horizontal lines to make the table more readable?
Table 2 It seems as there is one data row missing: vent – ranging- High outdoor
Table 3 ‘The relative organ weights or worm counts of free-range hens’. Consider to use ‘and’ instead of ‘or’ (2x)
Table 3 Replace ‘and’ by ‘,’ in ‘(indoor, low outdoor and high outdoor)’
Table 4 Replace ‘and’ by ‘,’ in ‘(indoor, low outdoor and high outdoor)’

---

## Round 0.2 · Minor Revisions

We would be grateful if you complete some minor suggestions made by reviewers'.

Reviewer 3 ·

Basic reporting

No comments.

Experimental design

No comments.

Validity of the findings

No comments.

Additional comments

Thank you for clarifying my questions.
I have only one remark left, concerning L226-228. To me as a rviewer it is now clear what you mean with 1.4h, but I am afraid that for a fresh reader the information between brackets in the manuscript now may be confusing. Consider to write the decimal times down as 1h 24min and 5h 12min. Or, if 1.4 is a standard way to write it down, change it back into the old phrasing.

The manuscript was not complete now; only table 3 was in it and no table captions at all.

---

## Round 0.3 · accepted · Accept

Thank you for having addressed the reviewers' comments. The manuscript is now ready for publication.